# Effects of Dietary Essential Oils Supplementation on Egg Quality, Biochemical Parameters, and Gut Microbiota of Late-Laying Hens

**DOI:** 10.3390/ani12192561

**Published:** 2022-09-25

**Authors:** Gengsheng Xiao, Liwei Zheng, Xia Yan, Li Gong, Yang Yang, Qien Qi, Xiangbin Zhang, Huihua Zhang

**Affiliations:** 1School of Life Science and Engineering, Foshan University, Foshan 528000, China; 2Guangdong Guangken Animal Husbandry Engineering Research Institute, Guangzhou 510000, China; 3Laboratory of Livestock and Poultry Breeding, Institute of Animal Science, Guangdong Academy of Agricultural Sciences, Guangzhou 510000, China; 4College of Animal Science, South China Agricultural University, Guangzhou 510642, China

**Keywords:** essential oils, laying hens, laying performance, egg quality, biochemical indices, microbiota

## Abstract

**Simple Summary:**

Improving the laying cycle of laying hens and improving the quality of eggs has always been the goal pursued by the laying hen industry. As a potential green and healthy feed additive, plant essential oil has important research significance. The results of this experiment suggest that plant essential oils may improve the egg quality of Dawu Golden Phoenix laying hens by affecting cecal flora and serum biochemical indexes. As far as we know, there are few studies on the effects of plant essential oils on cecal flora, serum biochemical indicators and their relationship with egg-laying performance.

**Abstract:**

The objective of this study was to explore the effects of adding essential oils (EO) to diets on egg quality, biochemical parameters and intestinal flora of late laying hens. The number of 252 Dawu Golden Phoenix laying hens (55 weeks old) were randomly sorted into two groups: the control group (CG) fed a basal diet and the EO group fed a basal diet with 300 mg/kg of essential oils. The average egg weight, feed-to-egg ratio, and egg production rate were determined every week. The trial started at week 55 and lasted for 8 weeks. During the experiment’s last week, 36 eggs out of each group were chosen at random to test. In our study, dietary supplementation with EO considerably decreased the egg breaking rate (*p* = 0.01) and increased the shell-breaking strength (*p* = 0.04). The treatment group’s alanine aminotransferase (ALT) levels were considerably lower than those of the control group (*p* = 0.03). The EO group had substantially higher total antioxidant capacity (T-AOC) and total superoxide dismutase (T-SOD) (*p* = 0.04 and *p* =0.03, respectively). However, there were no differences in alpha diversity indicators between the two groups. It is worth noting that *Firmicutes* were increased considerably (*p* < 0.05), while *Spirochaetota* and *Proteobacteria* were significantly reduced in the EO group. At genus levels, the EO supplementation increased the relative abundance of *Intestinimonas* (*p* < 0.05) and *Megamonas* (*p* < 0.01). In conclusion, a dietary supplementation of 300 mg/kg EO can improve the production performance of laying hens and the egg quality. It can also regulate the abundance of cecal flora and serum biochemical indicators.

## 1. Introduction

Eggs are a high-quality, nutritious protein food one needs on a daily basis [1] and are well-known as major providers of human nutrition, comprising proteins, lipids, vitamins, minerals and embryo development factors [2]. The cycle of commercial laying hens has been increased from 72 weeks to 80 weeks. After the peak egg-laying period, the number of follicles entering the preovulatory level of laying hens decreases, and the synthesis and accumulation of yolk also decreases, which ultimately leads to a decrease in the production performance [3]. Poor laying performance is indicated by a reduction in the laying rate, a drop in eggshell quality, and a rise in the feed-to-egg ratio [4]. Furthermore, lipid metabolic abnormalities, as well as visceral and abdominal fat deposition, develop with age, thereby impacting hen health and laying capacity [5]. However, using genetics and other forms of biotechnology to boost the productivity of laying hens is extremely challenging [6]. Therefore, it is necessary to find an effective feed additive to deal with this problem.

Essential oils (EO) (carvacrol, thyme and cinnamaldehyde) have garnered widespread interest due to the presence of bioactive compounds with antibacterial, antioxidant, and anti-inflammatory properties [7] and have provided a significant level of protection and enhancement in broiler productivity [8]. The presence of bioactive compounds (hydrocarbons, phenols, esters, alcohols, acids and steroids) in products of plant origin, which have positive effects on animal production and health [9]. One study found phytogenic plant extracts improved immunity in layer chickens, increased egg production and reduced cholesterol in serum and yolk [10]. In another study, dietary supplementation with plant essential oils significantly improved laying hen performance, egg production, ovarian morphology, serum lipid parameters and egg sensory quality [11,12]. With the gradual realization that plant extracts have natural antioxidant capacity and immune-promoting effects, the research on plant essential oils as feed supplements in laying hens’ performance has received more and more attention [13].

Maintaining a healthy level of intestinal development is essential to ensuring animals’ health and growth performance [14]. A variety of microorganisms reside in the chicken digestive tract, forming symbiotic relationships that affect nutrition, metabolism and immunity [15]. In numerous studies, plant essential oils are further claimed to improve intestinal health indices in poultry [16,17]. However, there are still relatively few studies about the effects of EO on the performance, serum biochemical indices and gut microbes of aged hens. This study aimed to investigate the effects of dietary EO on production, biochemical indicators and gut microflora during the late stage of hens.

## 2. Materials and Methods

### 2.1. Birds, Experimental Design and Diets

A total of 252 (55-week-old) Dawu Golden Phoenix laying hens were randomly divided into two groups (21 hens per replicate, 6 replicates per group). The control group (CG) were fed a basal diet and the EO group were fed a basal diet with 300 mg/kg of EO. The contents of the main constituents of the EO were as follows: carvacrol (2.5%), thyme (5%), cinnamaldehyde combined (2.5%) and carrier nanoscale silica (90%). The experiment started at week 55 and lasted for 8 weeks. The basal diet was formulated according to the nutrient requirements for laying hens (2012); the feed components and nutrient compositions are shown in Table 1. During the study, the birds were housed in metal cages (40 cm × 45 cm × 45 cm, Shandong Renhe Mechanical Equipment Co., Ltd., Taian, China) and shared a room that was cleaned and disinfected daily and was maintained at 26 ± 2 °C and 60–65% humidity with a 16 h light regime. Feed (100 g per hen per day) was provided at 7:00 a.m. and 2:00 p.m., and eggs were collected at 5:00 p.m. The birds had free access to drink water. Feeding, egg collection and the weighing of eggs were conducted daily.

### 2.2. Production and Egg Quality

Feed intake, number of eggs and egg breaking were recorded daily during a period of 55–62 wks. The egg-breaking rate was calculated based on the number of broken eggs. The egg production rate, feed-to-egg ratio, and average egg weight were calculated each week. During the last week of the experiment, 6 eggs were randomly selected from each replicate for a total of 36 eggs in each group. The eggs were used to determine Hough Units (Egg Analyzer, Orka Food Tech Ltd., Haifa, Israel), Shell Break Strength (Egg Force Reader, Orka Food Tech Ltd.) and Shell Thickness (Egg Shell Thickness Tester, Orka Foods Technology Co., Ltd.). The egg width-to-length ratio was calculated as the egg shape index (%).

### 2.3. Serum Biochemical Parameters and Intestinal Histomorphology

At the conclusion of the research, one bird was chosen randomly from each replication and blood was obtained following a 12 h period of feed deprivation. After 2 h of quiescence at 4 °C, blood samples were taken from the ulnar vein and the plasma was separated. Triglycerides, high density lipoprotein, low density lipoprotein, total cholesterol, total bilirubin, aspartate aminotransferase (AST) and ALT were all detected in serum using kits from Nanjing Jiancheng Bioengineering Institute (Nanjing, Jiangsu, China). The antioxidant parameters evaluated included MDA, T-AOC, T-SOD and glutathione peroxidase (GSH-PX), as per the Nanjing Jiancheng Bioengineering Institute kit’s instructions. Following the blood collection, the chosen hens were slaughtered and bled via cervical dislocation. The intestines of the left and right cecums were aseptically dissected out. One bird was randomly selected for each replicate, and a total of six euthanized birds were taken from each group. The jejunum and mid-ileum tissue samples were immersed in 4% formaldehyde solution for fixation. Then, the intestinal sample tissue was dehydrated and embedded in paraffin; 5 μm sections were cut serially, and 4 serial sections of each paraffin-embedded section were placed in xylene, rehydrated and stained with hematoxylin and eosin (H&E). The tissue sections were observed using a LEICA (DFC290 HD system digital camera, Heerbrugg, Switzerland) connected to an optical microscope with 10 objective lenses, and images were captured by Microsoft’s Image J processing software.

### 2.4. Microbial Analysis

At the conclusion of the investigation, the cecal microbiota of the chosen six chickens per group were assessed. After removing the cecal contents aseptically, the digesta was snap dried in liquid nitrogen, cooled to −80 °C, and the total genome DNA was extracted using a cetyltrimethylammonium bromide technique. The DNA content and purity were determined by 1.5% agarose gel electrophoresis. The amplification of the V3/V4 region of the 16S rRNA gene was achieved by using primers 341F (5′-CCTACGGGNGGCWGCAG-3′) and 804R (5′-GACTACHVGGGTATCTAATCC-3′). Fast Length Adjustment of Short Reads (V1.2.7) was used to merge paired-end reads into clean reads of high quality; QIIME (Quantitative Insight into Microbial Ecology) was used in a quality-control pipeline to filter the raw sequences and identify quality segments [18]. Clusters of high-quality sequences have been categorized as operational taxonomic units (OTUs) based on a similarity of 97 percent, and sample OTU sequences have been categorized as taxa using the SILVA database [19]. The two groups were visualized using a Venn diagram in order to show shared and distinct OTUs. The alpha diversity was calculated with the vegan package. For beta-diversity analysis, a principal coordinate analysis (PCoA) based on Bray–Curtis distance was used. STAMP with t-tests was used to examine differences in microbial abundances [20].

### 2.5. Statistical Analysis

All data were expressed as means ± SD using an independent samples t-test with SPSS 25.0 software (SPSS, Inc., Chicago, IL, USA) and presented using GraphPad Prism version 9 (GraphPad Software, La Jolla, CA, USA). A *p*-value < 0.05 was considered significant (*p* < 0.05, *p* < 0.01), and a 0.05 < *p*-value < 0.10 was discussed as a tendency.

## 3. Results

### 3.1. Production and Egg Quality

The values for egg weight, egg shape index, and Haugh unit did not significantly differ between the two groups in the current study. However, the EO group’s egg-breaking rate was lower than the control group’s (Table 2, *p* = 0.01), while the shell-breaking strength was significantly higher (Table 2, *p* = 0.04) in the EO group.

### 3.2. Serum Biochemical Indices and Intestinal Histomorphology

The ALT levels were lower in the treatment group than in the control group (Table 3, *p* = 0.03). The T-AOC and T-SOD of the EO group were considerably higher (Table 3, *p* < 0.05). AST and MDA showed a decreasing trend in the EO group (Table 3, *p* < 0.1)

### 3.3. Observation of Intestinal Histomorphology

The histomorphology of the intestinal segment is shown in Figure 1. The histological structure of the intestinal mucosa of laying hens was normal. Intestinal villi were broad and long in the jejunum and short and broad in the ileum. Observing the changes in the intestinal histomorphology between the two groups, the intestinal villi in the EO group were relatively more intact and healthier.

### 3.4. Microbial Composition of Cecal Contents

The two groups had a total of 2458 operational taxonomic units (OTUs), with 1054 and 373 OTUs discovered in the control and EO groups, respectively (Figure 2a). EO supplementation had no effect on the alpha diversity indices Ace, Chao1, Shannon, Simpson, goods coverage and PD whole tree (Table 4). The Bray–Curtis similarity method was used for PCOA analysis. Both the first principal component (PCOA1) and the second principal component (PCOA2) in turn contributed to the variation in microbial diversity by explaining 73.04% and 10% of it, respectively (Figure 2b). The samples in the control and EO groups were tightly grouped and not far apart on the primary coordinate analysis diagram, indicating that EO had no effect on the gut microbial communities in a specific direction. There are some changes in the structure of cecal flora after EO treatment, according to taxonomic research. Figure 2c,d displays the top 10 phyla and top 10 genera found in the cecal contents of the broilers. *Firmicutes, Bacteroidetes, Proteobacteria, Actinobacteria*, and *Verrucomicrobia* were the most prevalent phyla. The relative abundance of *Firmicutes* increased considerably in the EO group (Figure 2e, *p* < 0.05), whereas the *Spirochaetota* were significantly reduced (Figure 2e, *p* < 0.05). At the genus level, the dominant flora in cecal contents were *Bacteroides, Lactobacillus, Faecalibacterium,*
*Intestinimonas* and *Megamonas*. As a result of EO supplementation, the relative abundances of *Intestinimonas* and *Megamonas* were significantly increased (Figure 2e, *p* < 0.05).

## 4. Discussion

Plant essential oils such as carvacrol, thyme, and cinnamaldehyde serve multiple biochemical and physiological functions in the body, owing to their active components, a variety of antioxidant phytochemicals and bioactive ingredients [21]. In this study, the EO group showed a 1.08% increase in egg production and a significant increase in egg weight compared to the control group. Supplementation with an EO mixture including thymol and carvacrol to layer diets significantly increased egg weight and egg production compared to the control group [22]. Supplementation with 250 mg/kg thymol improved the performance of laying hens [23]. Adding plant-derived active substances to the diet can promote the digestion and absorption of nutrients in the gastrointestinal tract of broilers, thereby improving the growth performance of broilers [24]. In contrast, the growth performance of broilers was not affected by the inclusion of plant essential oils containing trans-anethole [25]. Notably, the average daily feed intake of White Leghorn broilers increased in a quadratic curve along with increasing levels of star anise oil (SAO) in the diet [26]. This report shows that the effects of different additive doses of EO on the body are different. The increase in egg weight and egg production in laying hens may be attributed to EO improving ovarian function and nutrient digestibility in the gut [27,28]. According to the literature, the positive effect of EO on the growth performance of broilers may be due to the bioactive substances of EO stimulating the secretion of digestive enzymes and pancreatic enzymes, thereby improving the intestinal morphology [29,30]. In our study, it was also found that the morphology of the intestinal villi in the EO group was more complete and longer than that in the control group.

In this study, supplementation with EO improved eggshell strength and eggshell thickness, reducing egg breakage. Essential Oils Premix supplementation has been reported to increase eggshell thickness and shell-breaking strength [22]. Improved shell-quality results with EO in feed have also been reported [31]. Similar results were reported in laying hens fed an EO diet of thyme and rosemary [32]. This is consistent with our results. The improved results obtained for eggshell quality indicators may be partly due to the effect of EO on the metabolic activity of beneficial colonies in the guts of laying hens positively affecting the absorption rate of minerals (especially Mg ^2+^ and Ca ^2+^) [33].

We all know that the antioxidant processes in an animal’s body are crucial for its health, growth, production and economic well-being. In this experiment, T-AOC and T-SOD levels for the EO group were also found to be significantly higher than those for the control group. A similar result was reported by S. Yarmohammadi Barbarestani, where a dietary supplement containing lavender essential oil increased the activity levels of T-SOD and T-AOC and decreased the concentration of MDA in the serum [34]. EO such as carvacrol and thymol possess antioxidant properties which are attributable to the presence of a phenolic hydroxyl group in their chemical structures acting as a hydrogen donor during the initial process of lipid oxidation in order to interact with peroxy radicals, thereby inhibiting the formation of hydrogen peroxide [35,36]. This shows that an EO (such as carvacrol, thyme and cinnamaldehyde) can be used as an exogenous antioxidant.

The existence of serum enzymes, as well as their levels in the blood, can help determine the extent of organ or tissue damage. ALT, AST and alkaline phosphatase enzymes are present in large quantities in liver cells. AST is located in the mitochondria, whereas ALT is located in the cytoplasm. These enzymes seep into the circulation as a result of the destruction of liver cells, which raises their concentration [37]. In this study, it was confirmed that EO played an important role in controlling liver function, consistent with the results of Ibrahim, D., et al. (2021) [38]. Sharma, Sharma and Kumar (2007) [39] found that peppermint extract inhibited the activity of liver enzymes in mice.

A growing body of evidence suggests that feed supplements have a dramatic effect on animal performance through the modulation of gut microbiota [20]. Recent studies have indicated the intestinal flora act as the main barrier against infection and colonization by pathogenic bacteria, implying the importance of their role in the treatment and prevention of diseases [40]. Our study showed that there was a certain difference in the relative abundance of certain intestinal flora between the EO group and the control group. Interestingly, a dietary intake of EO was able to reduce the levels of *Proteobacteria*, *Spirochaetota* and *Enterobacter* while increasing the relative abundance of *Firmicutes* and *Megamonas*.

*Firmicutes* facilitate cellulose digestion in the gut, so their higher abundance during growth and development aids in meeting animals’ nutritional and energetic needs [41]. Furthermore, *Firmicutes* are composed of a variety of gram-positive bacteria; some of these are thought to be probiotics that help to balance the gut flora and prevent the invasion of pathogens [42]. *Escherichia coli, Salmonella, Helicobacter pylori,* and *Vibrio cholerae* are the majority of the gram-negative bacteria that make up the proteobacteria as a whole. These bacteria provide a significant risk to human health, since they may cause gastrointestinal ulcers, vomiting, diarrhea, gastritis and even death in animals [43]. Adding a commercial phytogenic feed additive (a combination of 30 essential oils) to broiler diets increased LAB counts in cecal digesta. Other reports also showed that essential oils (thymol, eugenol, carvacrol and cinnamaldehyde) fortified the gut microflora by reducing harmful bacteria numbers and increasing beneficial bacteria populations [44,45]. 

*Megamonas* is a prominent member of the *Veillonellaceae* family. This genus uses fermentable fiber as a substrate to produce acetate and propionate in order to provide energy for the body, and this may also be the reason why the production performance of laying hens is affected by the intestinal flora [46]. The increased abundance of the *Firmicutes**, Megamonas* and *Intestinimonas* in the EO group found in this study may be affected by the phenolic active substances in EO. Through research, it was found that the phenolic active substances in EO can promote the colonization and growth of the *Megamonas* in the intestinal flora in the intestinal tract, thereby enhancing the intestinal absorption function [47]. This indicates that EO may improve gut microbiota ecology by increasing the relative abundance of *Firmicutes**, Megamonas* and *Intestinimonas* and reducing the *Proteobacteria* and *Spirochaetota*.

In conclusion, 300 mg/kg EO can effectively improve egg quality, serum biochemical indicators and intestinal flora ecology, but the control of the EO dose needs to be further refined. The specific mechanism underlying how *Firmicutes**, Megamonas*, *Intestinimonas, Proteobacteria* and *Spirochaetota* affect laying hens is worthy of further exploration.

## 5. Conclusions

This study found that adding 300 mg/kg EO to the diet could improve the eggshell quality and serum antioxidant capacity of laying hens in the later stage and reduce the egg-breaking rate. Although EO treatment did not affect the cecal flora composition and Alpha diversity index, it increased the abundance of *Firmicutes* (*p* < 0.05), *Intestinimonas* (*p* < 0.05) and *Megamonas* (*p* < 0.01) and decreased *Spirochaetota* (*p* < 0.05) and the abundance of *Proteobacteria*. This research has practical significance for the application of plant essential oils in laying hens, but the controls for the EO dose need to be further refined. The specific mechanism underlying how *Firmicutes**, Megamonas*, *Intestinimonas, Proteobacteria* and *Spirochaetota* affect laying hens is worthy of further exploration.

## Figures and Tables

**Figure 1 animals-12-02561-f001:**
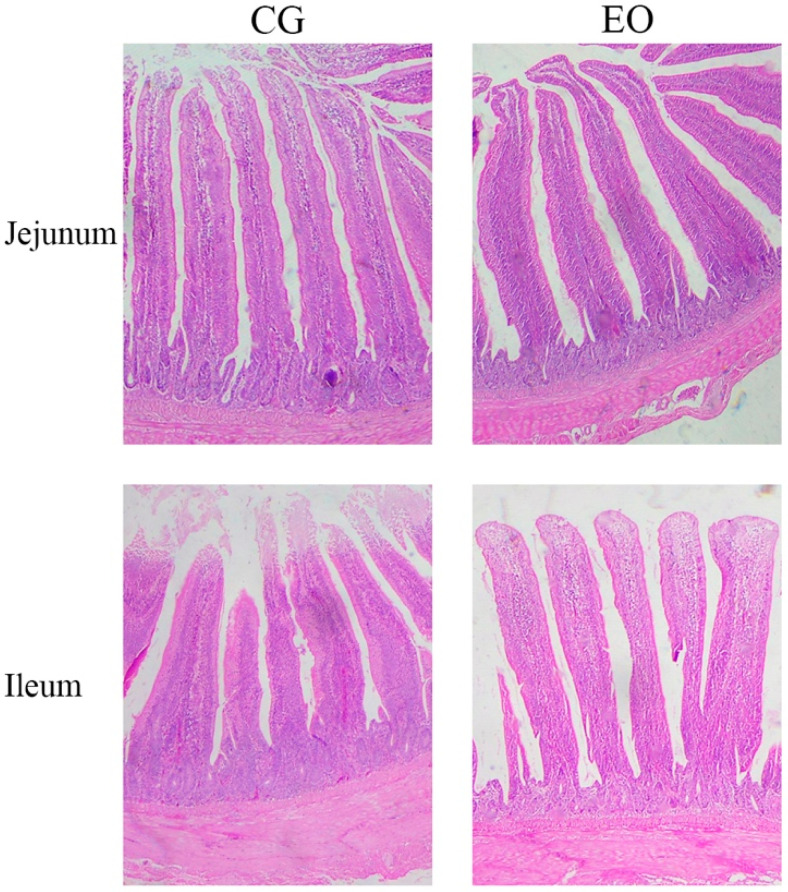
Jejunal and ileal mucosa photomicrograph with typical crypts and villi. The scale bar is 200 μm. Eosin and haematoxylin stain. Magnification was 100×. Abbreviations: CG, control group; EO, essential oils.

**Figure 2 animals-12-02561-f002:**
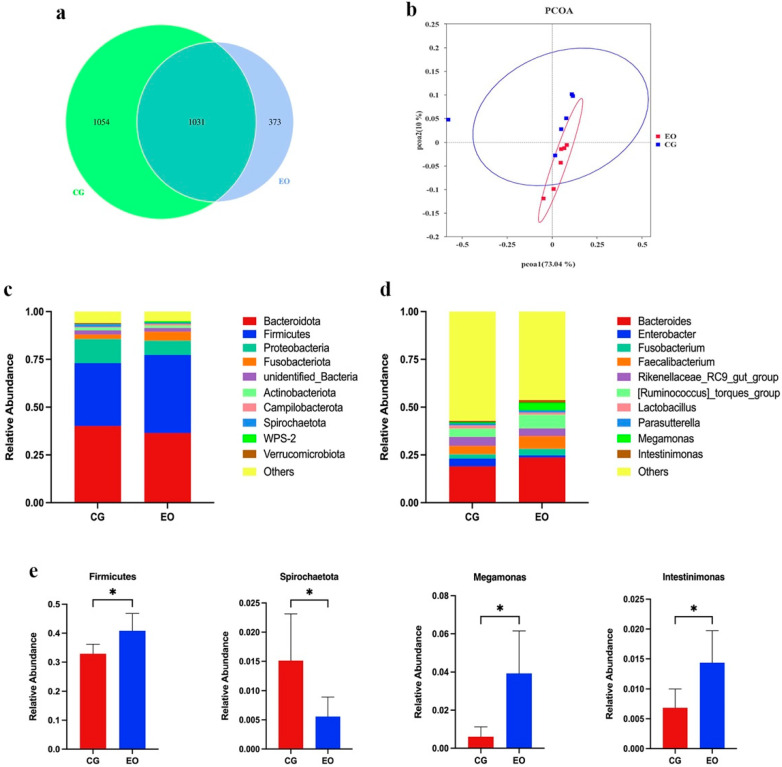
Microbial composition of cecal contents. (**a**) Venn diagram of number of operational taxonomic units (OTUs); (**b**) Principal coordinate analysis (PCOA) of the cecal microbiota; (**c**) Phylum-level taxonomic composition of the cecal microbiota or genus; (**d**) level; (**e**) Relative abundance of *Firmicutes**, Spirochaetota,*
*Intestinimonas,* and *Megamonas*. * Indicates statistically significant difference (*p* < 0.05). Abbreviations: CG, control group; EO, essential oils.

**Table 1 animals-12-02561-t001:** Basic diet’s ingredients and nutritional composition.

Feed Ingredients	%	Nutrient Composition	%
Corn	60.80	ME (kcal/kg)	4041.6
Soybean meal	26.00	CP	17.00
Linestone	7.74	Calcium	3.25
Soybean oil	2.62	Phosphorus	0.50
Calcium bicarbonate	1.40	Salt	0.03
Lysine	0.18	Lysine	0.998
DL-Methionine	0.18	DL-Methionine	0.435
Threonine	0.08		
1%Premix	1.00		
Total	100		

A percentage of 1% premix includes the following: vitamin A, 12,500 IU; vitamin D3, 4500 IU; vitamin E, 25 mg; vitamin B, 2 mg; vitamin K, 3 mg; vitamin B2, 30 mg; vitamin B12, 1 mg; niacin, 3 g; choline, 1500 mg; pantothenic acid, 700 mg; folic acid, 600 mg; Fe, 9 mg; Cu, 9 mg; Mn, 9 mg; I, 43 mg; Se, 30 mg; biotin, 0.2 mg.

**Table 2 animals-12-02561-t002:** Effects of EO supplementation on productive performance and egg quality.

Items	CG	EO	SEM	*p*-Value
Productive performance				
Egg production rate, %	81.62 ± 4.94	82.70 ± 3.59	1.08	0.64
Egg weight, g	60.80 ± 0.27	60.96 ± 0.89	0.84	0.35
Feed-to-egg ratio, g/g	2.01 ± 0.14	1.99 ± 0.12	0.03	0.80
Egg-breaking rate, %	0.65 ± 0.16	0.41 ± 0.13	0.36	0.01
Egg quality				
Egg shape index	1.31 ± 0.01	1.31 ± 0.01	0.01	0.89
Shell-breaking strength, kg/cm^2^	3.81 ± 0.08	3.98 ± 0.16	0.04	0.04
Shell thickness, mm	0.38 ± 0.02	0.39 ± 0.01	0.01	0.19
Haugh units	71.83 ± 7.34	74.02 ± 4.64	1.77	0.55

Abbreviations: CG, control group; EO, essential oils.

**Table 3 animals-12-02561-t003:** Effects of EO supplementation on serum biochemical indices and antioxidant parameters.

Items	CG	EO	SEM	*p*-Value
Serum biochemical indices				
Triglycerides, mmol/L	2.87 ± 0.64	2.75 ± 0.55	0.17	0.73
Total bilirubin, μmol/L	110.54 ± 14.53	96.20 ± 14.49	4.19	0.12
Total cholesterol, mmol/L	3.62 ± 1.06	5.03 ± 1.06	0.25	0.40
High density lipoprotein, mmol/L	5.18 ± 0.86	4.56 ± 1.02	0.27	0.28
Low density lipoprotein, mmol/L	0.32 ± 0.10	0.31 ± 0.15	0.04	0.83
Aspartate aminotransferase (AST), U/L	28.27 ± 6.68	20.98 ± 6.53	1.91	0.09
Alanine aminotransferase (ALT), U/L	12.85 ± 3.23	8.63 ± 2.62	0.85	0.03
Antioxidant parameters				
MDA, nmol/mL	5.65 ± 1.38	4.46 ± 0.55	0.30	0.08
T-AOC, U/mL	4.40 ± 1.68	6.32 ± 1.13	0.41	0.04
T-SOD, U/mL	75.04 ± 7.42	86.61 ± 8.80	2.35	0.03
GSH-PX, U/mL	594.36 ± 56.67	610.51 ± 62.56	17.21	0.65

Abbreviations: CG, control group; EO, essential oils; MDA, malondialdehyde; T-AOC, total antioxidant capacity; T-SOD, total superoxide dismutase; GSH-PH, glutathione peroxidase.

**Table 4 animals-12-02561-t004:** Effects of EO supplementation on the alpha diversity indices of cecal microbiota.

Items	CG	EO	SEM	*p*-Value
Ace	1012.25 ± 129.18	925.78 ± 132.83	37.82	0.28
Chao1	994.57 ± 115.81	917 ± 152.90	39.15	0.35
Shannon	6.84 ± 0.50	6.69 ± 0.36	0.13	0.56
Simpson	0.97 ± 0.03	0.97 ± 0.01	0.01	0.74
Goods_coverage	1.00 ± 0.01	1.00 ± 0.01	0.01	0.67
PD_whole tree	65.8 ± 10.09	56.58 ± 5.48	2.34	0.08

Abbreviations: CG, control group; EO, essential oils.

## Data Availability

The data presented in this study are available upon request from the corresponding author.

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
