# Peer review of "Effects of Dietary Essential Oils Supplementation on Egg Quality, Biochemical Parameters, and Gut Microbiota of Late-Laying Hens"

_animals, 2022, doi:10.3390/ani12192561_

Round 1

Reviewer 1 Report (Previous Reviewer 1)

I want to thank the authors for the changes made so far.

At the same time, once again, please explain what model it is and what is included in this model. Please explain the term: "fixed effects treatments" (Lines 145-146).

Author Response

Dear Editors and Reviewers,
Thank you for your letter and for the reviewers’ comments concerning our manuscript entitled “Effects of Dietary Essential Oils Supplementation on Egg Quality, Biochemical Parameters , and Gut Microbiota of Late Laying Hens” (ID: 1932251). Those comments are all valuable and very helpful for revising and improving our paper, as well as the important guiding significance to our researches. We have studied comments carefully and have made correction which we hope meet with approval. Revised portion are marked in red in the paper. The main corrections in the paper and the responds to the reviewer’s comments are as flowing:

Responds to the reviewer’s comments:
Comment 1: At the same time, once again, please explain what model it is and what is included in this model. Please explain the term: "fixed effects treatments" (Lines 145-146).

Response: Thank you for your professional advice. After we carefully browsed this sentence, we thought that the description was wrong, because we used the independent sample t test between the two groups for calculation, so we have deleted the sentence "fixed effects treatments".
      We did our best to improve the manuscript and made some revisions in the manuscript. These changes will not affect the content and framework of the paper. We are very grateful for the recognition and affirmation of the reviewers. We express our sincere gratitude to the editors and reviewers for their enthusiastic work, and hope that the revised manuscript will be recognized. Thanks again for your comments and suggestions.

Reviewer 2 Report (Previous Reviewer 2)

After the corrections were made, the manuscript is suitable for publishing because the discussion has improved greatly. I consider it to have good value for poultry production and is scientifically sound.

Author Response

Dear Editors and Reviewers,
       Thank you for your letter and for the reviewers’ comments concerning our manuscript entitled “Effects of Dietary Essential Oils Supplementation on Egg Quality, Biochemical Parameters , and Gut Microbiota of Late Laying Hens” (ID: 1932251). 

      Thank you very much for our revision to be recognized and affirmed by the reviewers! We sincerely thank the editors and reviewers for their enthusiastic work, and hope that the revised manuscript will be recognized and published. Thanks again for your comments and suggestions!

This manuscript is a resubmission of an earlier submission. The following is a list of the peer review reports and author responses from that submission.

Round 1

Reviewer 1 Report

The subject of the thesis, “Effects of Dietary Essential Oils Supplementation on Egg Quality, Biochemical Parameters, and Gut Microbiota of Late Laying Hens, is interesting, but the paper requires a thorough redrafting. 

General thoughts. 

Please check the work carefully and make changes so that the considerations in the manuscript will be consistent with the work's purpose (the Introduction section). 

Please describe in detail the materials and methods used in work. 

Selected detailed comments:

 Lines 47-77. Please make changes to the Introduction section and link them to the purpose of the work. 

Materials and Methods - This section is buggy, unclear, and needs fine-tuning. Please read and edit this passage carefully (concerns points 2.1-2.5). Authors must precisely describe the subject of the research and the research methodology. 

2.1. Birds, Experimental Design, and Diets 

Lines: 80-88. Please describe this passage in detail. The description of this passage is not clear. Which two groups of birds are there? How many birds are there in each group? What is included in the bird's base diet? What do essential oils mean? If the oils' composition is unknown, how can you analyze the results and then discuss the results? 

Lines 93-97. Please re-edit to describe the methodology in detail. It is unclear how many eggs were tested, what birds they came from, etc. 

Lines: 137-141. Please describe it in detail. What model is it about? What effects have been studied? Was the mean and standard deviation or error calculated?

 It is difficult to refer to the results of research and discussions due to the incomplete and imprecise description of materials and methods.

Author Response

Dear Editors and Reviewers,
Thank you for your letter and for the reviewers’ comments concerning our manuscript entitled “Effects of Dietary Essential Oils Supplementation on Egg Quality, Biochemical Parameters , and Gut Microbiota of Late Laying Hens” (ID: 1850930). Those comments are all valuable and very helpful for revising and improving our paper, as well as the important guiding significance to our researches. We have studied comments carefully and have made correction which we hope meet with approval. Revised portion are marked in red in the paper. The main corrections in the paper and the responds to the reviewer’s comments are as flowing:

Responds to the reviewer’s comments:
Comment 1: Please check the work carefully and make changes so that the considerations in the manuscript will be consistent with the work's purpose (the Introduction section). 
Response: Thank you for your professional advice. We have revised the introduction of the manuscript. Line 49-84.

Comment 2: Please describe in detail the materials and methods used in work.

Response: Thank you for your professional advice. We have revised the materials and methods of the manuscript. Line 85-241.

Comment 3: Lines 47-77. Please make changes to the Introduction section and link them to the purpose of the work. 

Response: Thank you for your professional advice. We have revised the introduction of the manuscript. Line 49-84.

Comment 4: Materials and Methods - This section is buggy, unclear, and needs fine-tuning. Please read and edit this passage carefully (concerns points 2.1-2.5). Authors must precisely describe the subject of the research and the research methodology. 

Response: Thank you for your professional advice. We have revised the materials and methods of the manuscript. Line 85-241.

Comment 5: 2.1. Birds, Experimental Design, and Diets. Lines: 80-88. Please describe this passage in detail. The description of this passage is not clear. Which two groups of birds are there? How many birds are there in each group? What is included in the bird's base diet? What do essential oils mean? If the oils' composition is unknown, how can you analyze the results and then discuss the results? 

Response: Thank you very much for your professional advice. We have revised in the manuscript. Line 85-156.

Comment 6: Lines 93-97. Please re-edit to describe the methodology in detail. It is unclear how many eggs were tested, what birds they came from, etc. 

Response: Thank you very much for your valuable advice. We have revised in the manuscript. Line 157-165.

Comment 7: Lines: 137-141. Please describe it in detail. What model is it about? What effects have been studied? Was the mean and standard deviation or error calculated?

Response: Thank you very much for your valuable advice. We have revised in the manuscript. Line 236-241.

Comment 8:  It is difficult to refer to the results of research and discussions due to the incomplete and imprecise description of materials and methods.

Response: Thank you for your professional advice. We have revised the materials and methods of the manuscript. Line 85-241.

Special thanks to you for your good comments. 

We tried our best to improve the manuscript and made some changes in the manuscript.  These changes will not influence the content and framework of the paper. And here we did not list the changes but marked in red in revised paper.
We appreciate for Editors and Reviewers’ warm work earnestly, and hope that the correction will meet with approval.
Once again, thank you very much for your comments and suggestions.

Reviewer 2 Report

The manuscript entitled “Effects of Dietary Essential Oils Supplementation on Egg Quality, Biochemical Parameters, and Gut Microbiota of Late Laying Hens” analyzes the use of 300 mg/Kg of plant essential oils added to the diet of laying hens and its effect on egg production, biochemical parameters, and the hens’ intestinal flora. Even though the aim of the research is interesting, it has several issues that need to be addressed before it can be published. It is well justified and the experimental design is adequate, but it needs a little bit of further explanation regarding which plants the essential oils (L82) come from and what is the reasoning for using these particular plants.

In L83, what do you mean by “the basic word was coined and used”? 

Also, if the hens were caged, how could they enjoy complete freedom of movement (L88)?  

In L97, clarify “egg primarily base”.

L102, do you mean the maxillary vein by saying zygomatic bone vein? Or which vein was used for sampling? And if this is correct, why wasn’t the more traditional ulnar vein chosen for sampling?

L112, this line should be in past explaining what was done, not written as if it was a set of instructions.

L185, in parenthesis you refer to figure 1e, but did you mean 2e?

The results obtained are quite interesting and extensive but are very poorly discussed. The discussion needs to be completely restructured for the manuscript to be considered for publishing, the possible reasons for the findings are not stated, and the authors do not try to explain their results, and how the essential oils could have affected, they limit the discussion to compare results which other studies. 

Why was egg production not affected? 

How did the different parameters evaluated affect the egg quality?

Why do the results suggest that the ability to overcome environmental stress may be more pronounced in the treated group, or how did the EO help the hens? 

Why were there some differences in the diversity of gut microbiota between the EO group and the control group, what could have happened?

Correlations occupy a big part of the results, and they are not discussed, either explain their importance in the study or remove them from the results.

And so forth, all the results should be discussed, giving a possible reason for why they happened or how. Don’t just compare your findings with other research.

Also, a final review of written English is recommended to correct any typos (L195) and mistakes.

Author Response

Dear Editors and Reviewers,
Thank you for your letter and for the reviewers’ comments concerning our manuscript entitled “Effects of Dietary Essential Oils Supplementation on Egg Quality, Biochemical Parameters , and Gut Microbiota of Late Laying Hens” (ID: 1850930). Those comments are all valuable and very helpful for revising and improving our paper, as well as the important guiding significance to our researches. We have studied comments carefully and have made correction which we hope meet with approval. Revised portion are marked in red in the paper. The main corrections in the paper and the responds to the reviewer’s comments are as flowing:

Responds to the reviewer’s comments:

Comment 1:  It is well justified and the experimental design is adequate, but it needs a little bit of further explanation regarding which plants the essential oils (L82) come from and what is the reasoning for using these particular plants.

Response: Thank you for your professional reminder. The contents of main constituents of EO were as follows: Carvacrol (2.5%), thyme (5%), cinnamaldehyde combined (2.5%) and carrier nanoscale silica (90%). Line 89-91. According to relevant literature Plant essential oils (carvacrol, thyme and cinnamaldehyde) garnered widespread interest due to the existence of bioactive compounds with antibacterial, antioxidant, and antiinflammatory properties and provided significant protection and enhancement in broiler productivity. Line 62-67.

Comment 2: In L83, what do you mean by “the basic word was coined and used”? Also, if the hens were caged, how could they enjoy complete freedom of movement (L88)?  

Response: Thank you very much for your valuable advice. We have revised in the manuscript. Line 15.

Comment 3: In L97, clarify “egg primarily base”.

Thank you very much for your professional advice. We have revised in the manuscript. Line 158-165.

Comment 4: L102, do you mean the maxillary vein by saying zygomatic bone vein? Or which vein was used for sampling? And if this is correct, why wasn’t the more traditional ulnar vein chosen for sampling?

Response: Thank you very much for your professional advice. An error occurred in our description. Blood samples were collected from the ulnar vein. We have revised in the manuscript. Line 169

Comment 5: L112, this line should be in past explaining what was done, not written as if it was a set of instructions.

Response: Thank you very much for your professional advice. We have revised in the manuscript. Line 176-180

Comment 6: L185, in parenthesis you refer to figure 1e, but did you mean 2e?

Response: Thank you very much for your valuable advice.  Yes, is figure 2e. We have revised in the manuscript. Line 302.

Comment 7: The results obtained are quite interesting and extensive but are very poorly discussed. The discussion needs to be completely restructured for the manuscript to be considered for publishing, the possible reasons for the findings are not stated, and the authors do not try to explain their results, and how the essential oils could have affected, they limit the discussion to compare results which other studies. 

Response: Thank you very much for your professional advice. We have revised the discussion of the manuscript. Line 93.

Comment 8: Why was egg production not affected? 

Response: Thank you very much for your professional advice. In this study, the egg production rate of the EO group was 1.08% higher than that of the control group. Although there is no statistically significant difference, this is the result of data obtained from production practice. The antioxidant, anti-inflammatory, and disease resistance of plant essential oils are relatively strong, which can also be consistent with the results of improved egg quality and intestinal flora, which is also very meaningful for actual production. We have revised in the manuscript. Line 334-347.

Comment 9: How did the different parameters evaluated affect the egg quality?

Response: Thank you very much for your professional advice. We have revised and elaborated in the discussion. Line 349-495.

Comment 10: Why do the results suggest that the ability to overcome environmental stress may be more pronounced in the treated group, or how did the EO help the hens? 

Response: Thank you very much for your valuable advice. We have revised and elaborated in the discussion. Line 355-369.

Comment 11: Why were there some differences in the diversity of gut microbiota between the EO group and the control group, what could have happened?

Correlations occupy a big part of the results, and they are not discussed, either explain their importance in the study or remove them from the results.

Response: Thank you very much for your professional advice. We have revised and elaborated in the discussion. Line 378-491.

Comment 12: And so forth, all the results should be discussed, giving a possible reason for why they happened or how. Don’t just compare your findings with other research.

Response: Thank you very much for your professional advice. We have revised in the manuscript. Line 491-496.

Comment 13: Also, a final review of written English is recommended to correct any typos (L195) and mistakes.

Response: Thank you very much for your patiently advice. We have revised in the manuscript.

Special thanks to you for your good comments. 

We tried our best to improve the manuscript and made some changes in the manuscript.  These changes will not influence the content and framework of the paper. And here we did not list the changes but marked in red in revised paper.
We appreciate for Editors and Reviewers’ warm work earnestly, and hope that the correction will meet with approval.
Once again, thank you very much for your comments and suggestions.

Round 2

Reviewer 1 Report

In the reviewer's opinion, the authors made only a slight correction without introducing significant changes.

The work has not been redrafted. Changing the order of the sentence is not an improvement when it comes to serious mistakes.

Therefore, the authors of the work did not address the reviewer's comments. The work still contains severe errors in the research methodology; it is chaotic.

Line 70: "attrq1zibuted"?

Line 106. Has the abbreviation EO been explained?

Author Response

Dear Editors and Reviewers,
Thank you for your letter and for the reviewers’ comments concerning our manuscript entitled “Effects of Dietary Essential Oils Supplementation on Egg Quality, Biochemical Parameters , and Gut Microbiota of Late Laying Hens” (ID: 1850930). Those comments are all valuable and very helpful for revising and improving our paper, as well as the important guiding significance to our researches. We have studied comments carefully and have made correction which we hope meet with approval. Revised portion are marked in red in the paper. The main corrections in the paper and the responds to the reviewer’s comments are as flowing:

Responds to the reviewer’s comments:
Reviewer 1: 
Comment 1: In the reviewer's opinion, the authors made only a slight correction without introducing significant changes.

The work has not been redrafted. Changing the order of the sentence is not an improvement when it comes to serious mistakes.

Therefore, the authors of the work did not address the reviewer's comments. The work still contains severe errors in the research methodology; it is chaotic.

Response: Thank you for your professional advice. You mentioned in the first-round review report that the materials and methods part of this manuscript is not clear, yes, you are right, in the first draft submission version our manuscript did have some inappropriate descriptions, but in the first round During the revision, we have tried our best to correct it. In this second round of revision, we have also made some revisions in the materials and methods section. We hope to gain your approval. We believe that the expression of this part of the content is clear and concise.

Comment 2: Line 70: "attrq1zibuted"?

Response: Thank you for your professional advice. We have revised the materials and methods of the manuscript. Line 66-68.

Comment 3: Line 106. Has the abbreviation EO been explained?

Response: Thank you for your professional advice. We have explained EO in the summary section and line 63.

Special thanks to you for your comments. 

We tried our best to improve the manuscript and made some changes in the manuscript.  And here we did not list the changes but marked in red in revised paper.We appreciate for Editors and Reviewers’ warm work earnestly, and hope that the correction will meet with approval.
Once again, thank you very much for your comments and suggestions.

Reviewer 2 Report

The manuscript has been improved, nonetheless, it is still necessary to improve the discussion by being more specific about the physiological and endocrinological processes triggered by the essential oils to improve the egg shell thickness. How do the essential oils work, and what do they do? Why do they enhance antioxidant capacity? Why does EO increase the abundance of Firmicutes and Megamonas? Even though this is mentioned in the discussion, it lacks a profound explanation of why the essential oils improve these parameters. 

Also, review typos found in L30, L58, L70, L81, L110 (reference).

Author Response

Dear Editors and Reviewers,
Thank you for your letter and for the reviewers’ comments concerning our manuscript entitled “Effects of Dietary Essential Oils Supplementation on Egg Quality, Biochemical Parameters , and Gut Microbiota of Late Laying Hens” (ID: 1850930). Those comments are all valuable and very helpful for revising and improving our paper, as well as the important guiding significance to our researches. We have studied comments carefully and have made correction which we hope meet with approval. Revised portion are marked in red in the paper. The main corrections in the paper and the responds to the reviewer’s comments are as flowing:

Responds to the reviewer’s comments:

Comment 1:  The manuscript has been improved, nonetheless, it is still necessary to improve the discussion by being more specific about the physiological and endocrinological processes triggered by the essential oils to improve the egg shell thickness. How do the essential oils work, and what do they do? Why do they enhance antioxidant capacity? Why does EO increase the abundance of Firmicutes and Megamonas? Even though this is mentioned in the discussion, it lacks a profound explanation of why the essential oils improve these parameters. 

Response: Thank you for your professional reminders. We have improved the Discussion section of the manuscript and hope to receive your approval. Line 348-515.

Comment 2: Also, review typos found in L30, L58, L70, L81, L110 (reference).

Response: Thank you very much for your valuable advice. We have revised in the manuscript. L30, L66-68, L70, L91.

Special thanks to you for your good comments. 

We tried our best to improve the manuscript and made some changes in the manuscript.  These changes will not influence the content and framework of the paper. And here we did not list the changes but marked in red in revised paper.
We appreciate for Editors and Reviewers’ warm work earnestly, and hope that the correction will meet with approval.
Once again, thank you very much for your comments and suggestions.